# E($n$)-Equivariant Spherical Decision Surfaces

**Pavlo Melnyk**[1]      **Michael Felsberg**[1]      **Kostas Daniilidis**[2]
[1]Linköping University      [2]University of Pennsylvania
pavlo.melnyk@liu.se

## ABSTRACT

We present a constructive derivation of exactly E($n$)-equivariant spherical decision surfaces by extending prior O($n$)-equivariant hypersphere neurons to include translations. To achieve this, we present a decomposition of the features of the O($n$)-equivariant neurons and provide explicit representations for translation and E($n$)-transformations to fulfil the respective equivariance constraints. The resulting decision surfaces are exactly E($n$)-equivariant without input centring or explicit pairwise differences, and admit explicit closed-form matrix representations. In addition, we numerically verify the correctness of the derivations and perform a downstream check of the resulting geometric primitives.

## 1 INTRODUCTION

Typically, O($n$)-equivariant[1] methods (Esteves et al., 2017; Deng et al., 2021; Assaad et al., 2023; Ruhe et al., 2023) require centring the input to become translation-equivariant, i.e., equivariant under the Euclidean group E($n$), or explicitly using the differences between input points, considering input as a collection of points (Thomas et al., 2018; Fuchs et al., 2020; Satorras et al., 2021). Motivated by the fact that spherical decision surfaces (Perwass et al., 2003) produce isometric activations, as shown by Melnyk et al. (2021) (which hints at the equivariance under the full Euclidean group), see Section 2.3, we aim to make O($n$)-equivariant hyperspheres (neurons) (Melnyk et al., 2024a) also translation-equivariant, beginning by considering only a single-point input.

Section 3 in its entirety constitutes the contributions of our work. Our goal is to establish and explicitly derive an exactly E($n$)-equivariant extension of O($n$)-equivariant spherical decision surfaces (Melnyk et al., 2024a) at the neuron level, rather than propose a composable deep architecture. We summarise our contributions as follows:

• We improve the understanding of the features learned by O($n$)-equivariant spherical neurons by showing how to decompose them into irreducible representations (irreps) — see Section 3.1;

• We derive the E($n$)-equivariant hypersphere neuron based on regular simplices and spherical decision surfaces (see (24), (26), and Figure 1), with the E($n$)-invariant primitives (41) and (43);

• We provide explicit transformation representations fulfilling the equivariance constraints (see (28) and (37)).

In addition, we numerically demonstrate the correctness of our derivations, with the results presented in Section 4. The code is available at *github.com/pavlo-melnyk/equivariant-hyperspheres*, under *notebooks/en_equiv_spherical_decision_surfaces/*.

## 2 BACKGROUND

In this section, we present a comprehensive background on the theory of spherical neurons (Perwass et al., 2003; Melnyk et al., 2021) and their rotation/reflection-equivariant version (Melnyk et al., 2022; 2024a), as well as on the general geometric concepts used in our work.

### 2.1 REGULAR SIMPLICES AND THE SIMPLEX CHANGE OF BASIS

Geometrically, a regular $n$-simplex represents $n + 1$ equidistant points in $n$D (Elte, 2006), lying on an $n$D sphere with unit radius. In the 2D case, the regular simplex is an equilateral triangle; in 3D, a regular tetrahedron, and so on.

---

[1]Here, we also include methods that are strictly speaking only SO($n$)-equivariant.

Following Cevikalp & Saribas (2023), we compute the Cartesian coordinates of a regular $n$-simplex as $n + 1$ vectors $\mathbf{p}_i \in \mathbb{R}^n$:

$$\mathbf{p}_i = \begin{cases} n^{-1/2}\,\mathbf{1}, & i = 1 \\ \kappa\,\mathbf{1} + \mu\,\mathbf{e}_{i-1}, & 2 \leq i \leq n + 1 \end{cases}, \quad \kappa = -\frac{1 + \sqrt{n+1}}{n^{3/2}}, \quad \mu = \sqrt{1 + \frac{1}{n}}, \tag{1}$$

where $\mathbf{1} \in \mathbb{R}^n$ is a vector with all elements equal to 1 and $\mathbf{e}_i$ is the natural basis vector with the $i$-th element equal to 1. For convenience we define $\boldsymbol{P}_n$ as an $n \times (n+1)$ matrix holding the regular $n$-simplex vertices, $\mathbf{p}_i$, in its columns:

$$\boldsymbol{P}_n = \begin{bmatrix} | & | & & | \\ \mathbf{p}_1 & \mathbf{p}_2 & \cdots & \mathbf{p}_n \\ | & | & & | \end{bmatrix}. \tag{2}$$

Based on the definition of the regular simplices (1), Melnyk et al. (2024a) generalised to $n$D the change-of-basis matrix defined in 3D by Melnyk et al. (2022) as an $(n+1) \times (n+1)$ orthogonal matrix $\boldsymbol{M}_n$ (i.e., representing rotation or reflection) holding in its columns the coordinates of the regular $n$-simplex appended with a constant and normalised to unit length:

$$\boldsymbol{M}_n = \left[\mathbf{m}_i\right]_{i=1}^{n+1}, \mathbf{m}_i = \frac{1}{p}\begin{bmatrix} \mathbf{p}_i \\ n^{-1/2} \end{bmatrix}, \quad p = \left\| \begin{bmatrix} \mathbf{p}_i \\ n^{-1/2} \end{bmatrix} \right\| = \sqrt{\frac{n+1}{n}}, \tag{3}$$

where the norms $p$ are constant, since $\|\mathbf{p}_i\| = \|\mathbf{p}_j\|$ for all $i$ and $j$ by definition of a regular simplex.

The following relation between $\boldsymbol{M}_n$ and $\boldsymbol{P}_n$ is important for the O($n$)-equivariance derivation of the neurons recapped in Section 2.4:

$$\boldsymbol{M}_n \boldsymbol{P}_n^\top = p \begin{bmatrix} \mathbf{I}_n \\ \mathbf{0}^\top \end{bmatrix}. \tag{4}$$

## 2.2   Equi- and Invariance under E($n$)-Transformations

The elements of the orthogonal group O($n$) can be represented as $n \times n$ matrices $\boldsymbol{R}$ with the properties $\boldsymbol{R}^\top \boldsymbol{R} = \boldsymbol{R}\boldsymbol{R}^\top = \mathbf{I}_n$, where $\mathbf{I}_n$ is the identity matrix, and $\det \boldsymbol{R} = \pm 1$, geometrically characterizing $n$D rotations and reflections. The special orthogonal group SO($n$) is a subgroup of O($n$) and includes only orthogonal matrices with the positive determinant, representing rotations.

The Euclidean group E($n$) is defined as a semidirect product E($n$) $\cong \mathbb{R}^n \rtimes$ O($n$); it consists of all orthogonal transformations together with translations, which can be represented as pairs $(\boldsymbol{R}, \mathbf{t})$: E($n$) $= \{(\boldsymbol{R}, \mathbf{t}) \mid \boldsymbol{R} \in$ O($n$)$, \ \mathbf{t} \in \mathbb{R}^n\}$ with $(\boldsymbol{R}_2, \mathbf{t}_2)(\boldsymbol{R}_1, \mathbf{t}_1) = (\boldsymbol{R}_2 \boldsymbol{R}_1, \ \mathbf{t}_2 + \boldsymbol{R}_2 \mathbf{t}_1)$.

We say that a function $f : \mathcal{X} \to \mathcal{Y}$ is E($n$)-equivariant if for every $\boldsymbol{R} \in$ O($n$) and $\mathbf{t} \in \mathbb{R}^n$, there exists the transformation representation $\rho(\boldsymbol{R}, \mathbf{t})$, in the function output space, $\mathcal{Y}$, such that

$$\rho(\boldsymbol{R}, \mathbf{t})\, f(\mathbf{x}) = f(\boldsymbol{R}\,\mathbf{x} + \mathbf{t}) \quad \text{for all } (\boldsymbol{R}, \mathbf{t}) \in \mathrm{E}(n), \ \mathbf{x} \in \mathcal{X} \subseteq \mathbb{R}^n, \tag{5}$$

where $\rho(\cdot)$ is a group representation, i.e., a homomorphism $\rho : \mathrm{E}(n) \to \mathrm{GL}(\mathcal{Y})$, mapping each $(\boldsymbol{R}, \mathbf{t})$ to an invertible linear operator on $\mathcal{Y}$ — a matrix operator in some chosen basis.

We call a function $f : \mathcal{X} \to \mathcal{Y}$ E($n$)-invariant if for every $\boldsymbol{R} \in$ O($n$) and $\mathbf{t} \in \mathbb{R}^n$, $\rho(\boldsymbol{R}, \mathbf{t}) = \mathbf{I}_n$. That is, if

$$f(\mathbf{x}) = f(\boldsymbol{R}\,\mathbf{x} + \mathbf{t}) \quad \text{for all } (\boldsymbol{R}, \mathbf{t}) \in \mathrm{E}(n), \ \mathbf{x} \in \mathcal{X} \subseteq \mathbb{R}^n. \tag{6}$$

Following the convention of prior work (Melnyk et al., 2022; 2024b) hereinafter, we write $\boldsymbol{R}$ to denote the same $n$D rotation/reflection as a $n \times n$ matrix in the Euclidean space $\mathbb{R}^n$, as a $(n+1) \times (n+1)$ matrix in the projective (homogeneous) space $P(\mathbb{R}^n) \subset \mathbb{R}^{n+1}$, and as an $(n+2) \times (n+2)$ matrix in $\mathbb{R}^{n+2}$. For the latter two cases, we achieve this by appending ones to the diagonal of the original $n \times n$ matrix without changing the transformation itself (Melnyk et al., 2021).

## 2.3   Spherical Neurons via Non-Linear Embedding

Spherical neurons (Perwass et al., 2003; Melnyk et al., 2021) are neurons with, as the name suggests, spherical decision surfaces.

**Sphere as a classifier** By virtue of conformal geometric algebra (Li et al., 2001), Perwass et al. (2003) proposed to embed the data vector $\mathbf{x} \in \mathbb{R}^n$ and represent the sphere with centre $\mathbf{c} = (c_1, \ldots, c_n) \in \mathbb{R}^n$ and radius $r \in \mathbb{R}$ respectively as

$$\boldsymbol{X} = \left(x_1, \ldots, x_n, -1, -\frac{1}{2}\|\mathbf{x}\|^2\right) \in \mathbb{R}^{n+2}, \qquad \boldsymbol{S} = \left(c_1, \ldots, c_n, \frac{1}{2}(\|\mathbf{c}\|^2 - r^2), 1\right) \in \mathbb{R}^{n+2}, \quad (7)$$

and used their scalar product $\boldsymbol{X}^\top \boldsymbol{S} = -\frac{1}{2}\|\mathbf{x} - \mathbf{c}\|^2 + \frac{1}{2}r^2$ as a classifier, i.e., the spherical neuron:

$$f_S(\boldsymbol{X}; \boldsymbol{S}) = \boldsymbol{X}^\top \boldsymbol{S}, \tag{8}$$

with learnable parameters $\boldsymbol{S} \in \mathbb{R}^{n+2}$. The sign of this scalar product depends on the position of the point $\mathbf{x}$ relative to the sphere $(\mathbf{c}, r)$: inside the sphere if positive, outside the sphere if negative, and on the sphere if zero (Perwass et al., 2003).

**Geometric interpretation and isometric activation** Geometrically, the activation of the spherical neuron (8) determines the cathetus length of the right triangle formed by $\mathbf{x}$, $\mathbf{c}$, and the corresponding point on the sphere (see Figure 2 in Melnyk et al. (2021)), and, as shown by Melnyk et al. (2021), is *isometric*, i.e., rigid transformations of the spherical neuron (or rather, its spherical decision surface) correspond to the rigid transformations of the vector it acts upon: Using two $(n+2) \times (n+2)$ *motor*, i.e., rigid transformation, operators $\boldsymbol{Q}_S$ and $\boldsymbol{Q}_X$ that act on $\boldsymbol{S}$ and $\boldsymbol{X}$ (7), respectively,

$$\boldsymbol{Q}_S = \begin{bmatrix} \boldsymbol{R} & \mathbf{0} & \mathbf{t} \\ \mathbf{t}^\top \boldsymbol{R} & 1 & \frac{1}{2}\|\mathbf{t}\|^2 \\ \mathbf{0}^\top & 0 & 1 \end{bmatrix} \quad \text{and} \quad \boldsymbol{Q}_X = \begin{bmatrix} \boldsymbol{R} & -\mathbf{t} & \mathbf{0} \\ \mathbf{0}^\top & 1 & 0 \\ -\mathbf{t}^\top \boldsymbol{R} & \frac{1}{2}\|\mathbf{t}\|^2 & 1 \end{bmatrix}, \tag{9}$$

such that $\boldsymbol{Q}_X$ is the adjoint of $\boldsymbol{Q}_S$, i.e., $\boldsymbol{Q}_X^\top \boldsymbol{Q}_S = \mathbf{I}_{n+2}$, we obtain

$$(\boldsymbol{Q}_X \boldsymbol{X})^\top (\boldsymbol{Q}_S \boldsymbol{S}) = \boldsymbol{X}^\top \boldsymbol{Q}_X^\top \boldsymbol{Q}_S \boldsymbol{S} = \boldsymbol{X}^\top \boldsymbol{S}. \tag{10}$$

**Non-linear activation** We note that with respect to the data vector $\mathbf{x} \in \mathbb{R}^n$, a spherical neuron represents a non-linear function $f_S(\,\cdot\,; \boldsymbol{S}) : \mathbb{R}^{n+2} \to \mathbb{R}$, due to the inherent non-linearity of the embedding (7), and therefore, does not necessarily require an activation function, as observed by Melnyk et al. (2021).

**Independent learnable parameters** The components of $\boldsymbol{S}$ in (7) can be treated as *independent* learnable parameters. In this case, a spherical neuron learns a *non-normalised* sphere of the form $\widetilde{\boldsymbol{S}} = (\widetilde{s}_1, \ldots, \widetilde{s}_{n+2}) \in \mathbb{R}^{n+2}$, which represents the same decision surface as its normalised counterpart defined in (7), thanks to the homogeneity of the embedding (Perwass et al., 2003; Li et al., 2001). The last parameter of $\widetilde{\boldsymbol{S}}$ is called the scale parameter $\gamma := \widetilde{s}_{n+2}$, so the definition (8) can be written more generally as

$$f_S(\boldsymbol{X}; \widetilde{\boldsymbol{S}}) = \gamma \, \boldsymbol{X}^\top \boldsymbol{S}. \tag{11}$$

For simplicity and without loss of generality, in the following we suppose $\gamma = 1$.

## 2.4 O($n$)-EQUIVARIANT HYPERSPHERES (NEURONS)

Following the proposal for 3D by Melnyk et al. (2022), which was later utilised in an end-to-end method (Melnyk et al., 2024b), Melnyk et al. (2024a) generalised to $n$D an O($n$)-equivariant hypersphere neuron (also referred to as *equivariant hypersphere*) based on spherical decision surfaces (Perwass et al., 2003; Melnyk et al., 2021) and regular simplices (see Section 2.1).

**Definition** Particularly, an equivariant hypersphere $\mathbf{F}_n(\boldsymbol{X}; \boldsymbol{S})$ is defined as the $(n+1) \times (n+2)$ matrix $(\boldsymbol{S})$ for the spherical decision surface $\boldsymbol{S} \in \mathbb{R}^{n+2}$ with centre $\mathbf{c} \in \mathbb{R}^n$ and an $n$D input $\mathbf{x} \in \mathbb{R}^n$ embedded as $\boldsymbol{X} \in \mathbb{R}^{n+2}$ as

$$\mathbf{F}_n(\boldsymbol{X}; \boldsymbol{S}) = B(\boldsymbol{S}) \, \boldsymbol{X}, \quad B(\boldsymbol{S}) = \left[(\boldsymbol{R}_O^\top \, \boldsymbol{R}_{T_i} \, \boldsymbol{R}_O \, \boldsymbol{S})^\top\right]_{i=1}^{n+1}, \tag{12}$$

where $\{\boldsymbol{R}_{T_i}\}_{i=1}^{n+1}$ is the $\mathbb{R}^{n+2}$ rotation isomorphism corresponding to the rotation from the first vertex to the $i$-th vertex of the regular $n$-simplex (1) (and therefore, $\boldsymbol{R}_{T_1} = \mathbf{I}_{n+2}$), and $\boldsymbol{R}_O \in \mathrm{SO}(n)$ is the

geodesic (shortest) rotation[2] from the sphere centre $\mathbf{c}$ to $\|\mathbf{c}\|\,\mathbf{p}_1$:

$$\boldsymbol{R}_O\,\mathbf{c} = \|\mathbf{c}\|\mathbf{p}_1\,, \tag{13}$$

where $\mathbf{p}_1 \in \mathbb{R}^n$ is the first vertex of the regular simplex according to (1). Hence, $\boldsymbol{R}_{T_i}\,\boldsymbol{R}_O\,\mathbf{c} = \|\mathbf{c}\|\mathbf{p}_i$. Note that if the centre $\mathbf{c}$ happens to be $-\mathbf{p}_1$, $\boldsymbol{R}_O$ is a reflection about the origin. Thus, in principle, we could also write $\boldsymbol{R}_O \in \mathrm{O}(n)$.

The definition of $B(\boldsymbol{S})$ in (12) can be written by using the definition of a spherical decision surface (7) and a regular simplex (1) and (2), as well as the definition of $\boldsymbol{R}_O$ (13):

$$
\begin{aligned}
B(\boldsymbol{S}) &= \left[(\boldsymbol{R}_O^\top\,\boldsymbol{R}_{T_i}\,\boldsymbol{R}_O\,\boldsymbol{S})^\top\right]_{i=1}^{n+1} = \left[\mathbf{c}^\top\,\boldsymbol{R}_O^\top\,\boldsymbol{R}_{T_i}^\top\,\boldsymbol{R}_O \quad s_{n+1} \quad 1\right]_{i=1}^{n+1} \\
&= \left[\|\mathbf{c}\|\mathbf{p}_i^\top\,\boldsymbol{R}_O \quad s_{n+1} \quad 1\right]_{i=1}^{n+1} = \left[\|\mathbf{c}\|\,\boldsymbol{P}_n^\top\,\boldsymbol{R}_O \quad s_{n+1}\mathbf{1} \quad \mathbf{1}\right],
\end{aligned}
\tag{14}
$$

where $s_{n+1} := \frac{1}{2}(\|\mathbf{c}\|^2 - r^2)$.

**$\mathrm{O}(n)$-equivariance**    The $\mathrm{O}(n)$-equivariance property for the equivariant hypersphere is

$$\boldsymbol{V}(\boldsymbol{R})\,B(\boldsymbol{S})\,\boldsymbol{X} = B(\boldsymbol{S})\,\boldsymbol{R}\boldsymbol{X}, \tag{15}$$

where $\boldsymbol{R}$ is the rotation representation in $\mathbb{R}^{n+2}$ obtained by extending the diagonal of the original $n \times n$ matrix with 1s (ones), and the transformation representation in the neuron output space $\mathbb{R}^{n+1}$ is given by the $(n+1) \times (n+1)$ matrix

$$\boldsymbol{V}(\boldsymbol{R}) := \boldsymbol{M}_n^\top\,\boldsymbol{R}_O\,\boldsymbol{R}\,\boldsymbol{R}_O^\top\boldsymbol{M}_n\,, \tag{16}$$

where $\boldsymbol{M}_n \in \mathrm{O}(n+1)$ is the-change-of-basis matrix defined in (3) and a 1 is appended to the diagonals of $\boldsymbol{R}_O$ and $\boldsymbol{R}$ to make them $(n+1) \times (n+1)$. Furthermore, $\boldsymbol{V}(\boldsymbol{R})$ belongs to a proper subgroup of $\mathrm{O}(n+1)$: $G = \mathrm{O}(n) \times \mathcal{S}^n$, i.e., $G < \mathrm{O}(n+1)$, where $G$ is formed as $\mathrm{O}(n) \times \boldsymbol{M}_n\left(\frac{\mathbf{c}}{\|\mathbf{c}\|}, 0\right)$ with $\boldsymbol{M}_n\left(\frac{\mathbf{c}}{\|\mathbf{c}\|}, 0\right) \in \mathcal{S}^n$.

The original transformation is found directly as

$$\boldsymbol{R} = \boldsymbol{R}_O^\top\boldsymbol{M}_n\,\boldsymbol{V}(\boldsymbol{R})\,\boldsymbol{M}_n^\top\boldsymbol{R}_O\,, \tag{17}$$

followed by the retrieval of the upper-left $n \times n$ sub-matrix.

**Non-linearity**    With respect to the input vector $\mathbf{x} \in \mathbb{R}^n$, the equivariant hypersphere $\mathbf{F}_n(\,\cdot\,;\boldsymbol{S}) : \mathbb{R}^{n+2} \to \mathbb{R}^{n+1}$ represents a non-linear $\mathrm{O}(n)$-equivariant function. Also, the *sum* of the output $B(\boldsymbol{S})\,\boldsymbol{X}$ is an $\mathrm{O}(n)$-invariant scalar due to the regular $n$-simplex construction.

## 3    THEORETICAL RESULTS

In this section, we present the main results of our work. Complete derivation and proofs are presented in Appendix A, which, for convenience, follows the same order as this section.

For the derivation of translation equivariance, it is instrumental first to consider the following decomposition of the output $\mathbf{y}$ of the $\mathrm{O}(n)$-equivariant hypersphere neuron.

### 3.1    IDENTITY SPLIT: EQUIVARIANT HYPERSPHERE OUTPUT DECOMPOSITION INTO IRREPS

Given $\boldsymbol{X} \in \mathbb{R}^{n+2}$ (7) and $\mathbf{y} := B(\boldsymbol{S})\,\boldsymbol{X} \in \mathbb{R}^{n+1}$, we decompose $\mathbf{y}$ into $\mathrm{O}(n)$-equivariant and invariant parts using (14) as

$$\boldsymbol{R}_O^\top\,\boldsymbol{M}_n\,\mathbf{y} = \boldsymbol{R}_O^\top\boldsymbol{M}_n\,B(\boldsymbol{S})\,\boldsymbol{X} = p\,\|\mathbf{c}\|\begin{bmatrix}\mathbf{x} \\ \alpha\,(r^2 - \|\mathbf{c}\|^2 - \|\mathbf{x}\|^2)\end{bmatrix}, \tag{18}$$

---

[2]In practice, it is computed utilising the Householder (double-) reflection method, e.g., as described by Golub & Van Loan (2013).

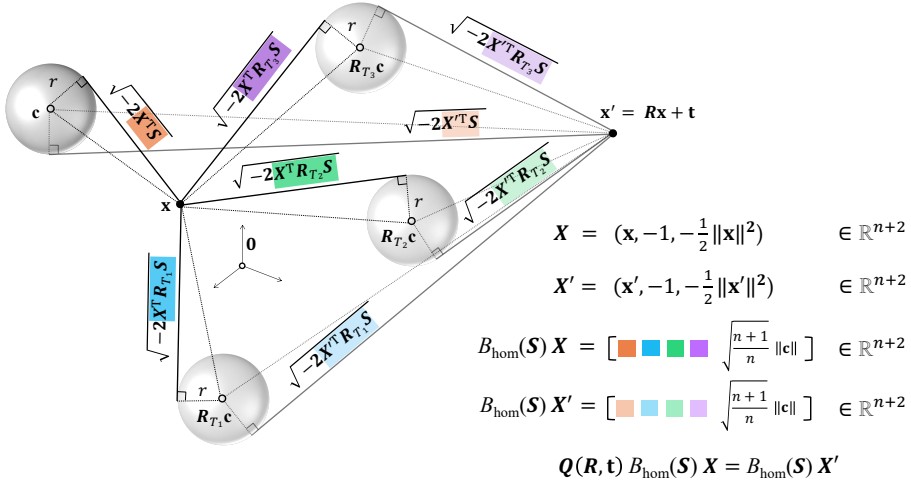

$$X = (\mathbf{x}, -1, -\tfrac{1}{2}\|\mathbf{x}\|^2) \quad \in \mathbb{R}^{n+2}$$

$$X' = (\mathbf{x}', -1, -\tfrac{1}{2}\|\mathbf{x}'\|^2) \quad \in \mathbb{R}^{n+2}$$

$$B_{\mathrm{hom}}(S)\,X = \left[\ \blacksquare\ \blacksquare\ \blacksquare\ \blacksquare\ \sqrt{\tfrac{n+1}{n}}\ \|\mathbf{c}\|\ \right] \quad \in \mathbb{R}^{n+2}$$

$$B_{\mathrm{hom}}(S)\,X' = \left[\ \blacksquare\ \blacksquare\ \blacksquare\ \blacksquare\ \sqrt{\tfrac{n+1}{n}}\ \|\mathbf{c}\|\ \right] \quad \in \mathbb{R}^{n+2}$$

$$Q(R, \mathbf{t})\,B_{\mathrm{hom}}(S)\,X = B_{\mathrm{hom}}(S)\,X'$$

Figure 1: **Geometric interpretation of the equivariant hypersphere neuron and its homogeneous extension**, $B_{\mathrm{hom}}(S)$ (best viewed in colour). Each component of $B(S)\,X$ corresponds to the scalar product between the embedded input $X = (\mathbf{x}, -1, -\tfrac{1}{2}\|\mathbf{x}\|^2)$ and a spherical decision surface transformed by the simplex rotations $R_{T_i}$. The homogeneous operator $B_{\mathrm{hom}}(S)$ augments the output with an additional coordinate depending on the space dimensionality $n$ and the sphere centre norm, $\|\mathbf{c}\|$, enabling a linear representation $Q(R, \mathbf{t})$ of the Euclidean group $\mathrm{E}(n)$. Only the parameters defining the sphere $S$—its centre $\mathbf{c}$ and radius $r$—are learnable; the remaining quantities arise from the geometric construction.

where $\alpha := \frac{\sqrt{n+1}}{2p\,\|\mathbf{c}\|} = \frac{\sqrt{n}}{2\,\|\mathbf{c}\|}$, and the invariant part is $\frac{\sqrt{n+1}}{2}\left(r^2 - \|\mathbf{c}\|^2 - \|\mathbf{x}\|^2\right)$, with $\mathbf{c}$ and $r$ being learnable parameters.

Observe that in the case $\|\mathbf{c}\| = 0$, the $\mathrm{O}(n)$-equivariant part, i.e., the original vector $\mathbf{x}$, zeroes out, and what we are left with is

$$R_O^\top\,M_n\,\mathbf{y} = R_O^\top M_n\,B(S)\,X = \begin{bmatrix} \mathbf{0} \\ \frac{p\sqrt{n}}{2}\left(r^2 - \|\mathbf{x}\|^2\right) \end{bmatrix}. \tag{19}$$

Also, if $X$ is transformed with $R \in \mathrm{O}(n)$ extended to $(n+2) \times (n+2)$ using the identity matrix, as $R\,X$, defining $\mathbf{y}_R := B(S)\,R\,X$, we obtain

$$R_O^\top\,M_n\,\mathbf{y}_R = R_O^\top M_n\,B(S)\,R\,X = p\,\|\mathbf{c}\| \begin{bmatrix} R\,\mathbf{x} \\ \alpha\left(r^2 - \|\mathbf{c}\|^2 - \|\mathbf{x}\|^2\right) \end{bmatrix}. \tag{20}$$

### 3.2 Translation Equivariance

The desired equivariance property is

$$T(\mathbf{t})\,B(S)\,X = B(S)\,T_{\mathrm{RHS}}\,X, \tag{21}$$

where $T(\mathbf{t})$ is the translation representation we seek and

$$T_{\mathrm{RHS}} = \begin{bmatrix} \mathbf{I}_n & -\mathbf{t} & \mathbf{0} \\ \mathbf{0}^\top & 1 & 0 \\ -\mathbf{t}^\top & \tfrac{1}{2}\|\mathbf{t}\|^2 & 1 \end{bmatrix} \tag{22}$$

is the $(n+2) \times (n+2)$ matrix (9) with $R = \mathbf{I}_n$.

Consider the RHS and denote $\mathbf{y_t}$ to be the output of the equivariant hypersphere given the input $\mathbf{x}$ translated as $\mathbf{x} + \mathbf{t}$ and embedded in $\mathbb{R}^{n+2}$ (see (7)) as $X_\mathbf{t}$:

$$\mathbf{y_t} := B(\boldsymbol{S})\,\boldsymbol{X}_t = B(\boldsymbol{S})\,\boldsymbol{T}_{\text{RHS}}(\mathbf{t})\,\boldsymbol{X} \ \in \mathbb{R}^{n+1}. \tag{23}$$

As we show in the Appendix, to satisfy (21), we need to make the original $(n+1)\times(n+2)$ equivariant hypersphere $B(\boldsymbol{S})$ (12) output a *homogeneous* feature. For this, we create the $(n+2)\times(n+2)$ $B_{\text{hom}}(\boldsymbol{S})$:

$$B_{\text{hom}}(\boldsymbol{S}) := \begin{bmatrix} & B(\boldsymbol{S}) & \\ \mathbf{0}^\top & -p\,\|\mathbf{c}\| & 0 \end{bmatrix}, \tag{24}$$

so that

$$B_{\text{hom}}(\boldsymbol{S})\,\boldsymbol{X} = \begin{bmatrix} B(\boldsymbol{S})\,\boldsymbol{X} \\ p\,\|\mathbf{c}\| \end{bmatrix} = \begin{bmatrix} \mathbf{y} \\ p\,\|\mathbf{c}\| \end{bmatrix} \in \mathbb{R}^{n+2}, \tag{25}$$

where $\mathbf{c} \in \mathbb{R}^n$ are parameters in $\boldsymbol{S}$, and hence no new learnable parameters are added, and the $-$ sign in the $-p\,\|\mathbf{c}\|$ element in $B_{\text{hom}}(\boldsymbol{S})$ is due to the second last element of the embedding $\boldsymbol{X}$ (7) being $-1$.

Observe that with respect to the input vector $\mathbf{x} \in \mathbb{R}^n$, the equivariant hypersphere (24) $\mathbf{F}_n(\,\cdot\,;\boldsymbol{S}) : \mathbb{R}^{n+2} \to \mathbb{R}^{n+2}$ represents a non-linear $\mathrm{E}(n)$-equivariant function. In the general case with the learnable $\gamma \in \mathbb{R}$ parameter (see Section 2.3), we define $B_{\text{hom}}(\widetilde{\boldsymbol{S}})$ as

$$B_{\text{hom}}(\widetilde{\boldsymbol{S}}) := \gamma \begin{bmatrix} & B(\boldsymbol{S}) & \\ \mathbf{0}^\top & -p\,\|\mathbf{c}\| & 0 \end{bmatrix}. \tag{26}$$

We rewrite the desired equivariance property (21) for the new $B_{\text{hom}}(\boldsymbol{S})$:

$$\boldsymbol{T}(\mathbf{t})\,B_{\text{hom}}(\boldsymbol{S})\,\boldsymbol{X} = B_{\text{hom}}(\boldsymbol{S})\,\boldsymbol{T}_{\text{RHS}}\,\boldsymbol{X}, \tag{27}$$

where we find that

$$\boldsymbol{T}(\mathbf{t}) := \boldsymbol{M}_n^\top\,\boldsymbol{R}_O\,\boldsymbol{T}_{\text{interm}}\,\boldsymbol{R}_O^\top\,\boldsymbol{M}_n, \tag{28}$$

where, gently abusing notation, $\boldsymbol{M}_n \in \mathrm{O}(n+1)$ and $\boldsymbol{R}_O \in \mathrm{O}(n)$ are extended to $(n+2)\times(n+2)$ using the identity matrix, and $\boldsymbol{T}_{\text{interm}}$ is the $(n+2)\times(n+2)$ matrix

$$\boldsymbol{T}_{\text{interm}}(\mathbf{t}) = \begin{bmatrix} \mathbf{I}_n & 0 & \mathbf{t} \\ -2\alpha\,\mathbf{t}^\top & 1 & -\alpha\|\mathbf{t}\|^2 \\ \mathbf{0}^\top & 0 & 1 \end{bmatrix}, \tag{29}$$

where $\alpha = \frac{\sqrt{n+1}}{2p\,\|\mathbf{c}\|} = \frac{\sqrt{n}}{2\,\|\mathbf{c}\|}$.

We verify that $\boldsymbol{T}(\mathbf{t})$ is indeed a linear representation as necessary for the equivariance definition (5), by (a) finding its inverse

$$\boldsymbol{T}(\mathbf{t})^{-1} = \boldsymbol{M}_n^\top\,\boldsymbol{R}_O\,\boldsymbol{T}_{\text{interm}}^{-1}\,\boldsymbol{R}_O^\top\,\boldsymbol{M}_n, \quad \boldsymbol{T}_{\text{interm}}^{-1} = \begin{bmatrix} \mathbf{I}_n & 0 & -\mathbf{t} \\ 2\alpha\,\mathbf{t}^\top & 1 & -\alpha\|\mathbf{t}\|^2 \\ \mathbf{0}^\top & 0 & 1 \end{bmatrix}, \tag{30}$$

which is valid for all $\mathbf{t} \in \mathbb{R}^n$ and $\alpha$ with $\|\mathbf{c}\| \neq 0$ (we will consider this case below), and (b) verifying that the action of $\boldsymbol{T}(\mathbf{t}_2)\,\boldsymbol{T}(\mathbf{t}_1)$ results in the translation of the original $n\mathrm{D}$ $\mathbf{x}$ as $\mathbf{x} + \mathbf{t}_1 + \mathbf{t}_2$:

$$\boldsymbol{T}(\mathbf{t}_2)\,\boldsymbol{T}(\mathbf{t}_1) = \boldsymbol{M}_n^\top\,\boldsymbol{R}_O\,\boldsymbol{T}_{\text{interm}}(\mathbf{t}_1 + \mathbf{t}_2)\,\boldsymbol{R}_O^\top\,\boldsymbol{M}_n \ . \quad \square \tag{31}$$

**Degenerate case $\mathbf{c} = \mathbf{0}$** Geometrically speaking, the case $\mathbf{c} = \mathbf{0}$ means that the entire regular simplex construction in $B(\boldsymbol{S})$ (14) collapses to a single sphere with three identical copies, all centred at $\mathbf{0}$; additionally, it means that $\boldsymbol{R}_O$ (13) does not exist. In practice, we can prevent the degenerate case, e.g., by adding some negligible $\epsilon$ to $\|\mathbf{c}\|$.

### 3.3 Completing to $\mathrm{E}(n)$-Equivariance

The desired equivariance property is

$$\boldsymbol{Q}(\boldsymbol{R}, \mathbf{t})\,B_{\text{hom}}(\boldsymbol{S})\,\boldsymbol{X} = B_{\text{hom}}(\boldsymbol{S})\,\boldsymbol{Q}_{\text{RHS}}\,\boldsymbol{X}, \tag{32}$$

where $\boldsymbol{Q}_{\text{RHS}} := \boldsymbol{Q}_X$ (9).

By using an identity complement and extending $V$ (16) to $(n+2) \times (n+2)$ on the LHS in (15), we note that $B_{\text{hom}}(S)$ (24) is also $\mathrm{O}(n)$-equivariant: since the RHS in (15) can be rewritten as

$$B_{\text{hom}}(S) \, \boldsymbol{R} \, X = \begin{bmatrix} B(S)\,\boldsymbol{R}\,X \\ p\|\mathbf{c}\| \end{bmatrix}, \tag{33}$$

where $p\|\mathbf{c}\| = \sqrt{\frac{n+1}{n}}\,\|\mathbf{c}\|$ and $\|\mathbf{x}\|^2 = \|\boldsymbol{R}\,\mathbf{x}\|^2$ are $\mathrm{O}(n)$-invariant. The same is valid for the LHS, where the additional $(n+2)$-th element in the resulting vector is also $p\|\mathbf{c}\|$.

In turn, the RHS of (32) is

$$\mathbf{y}_{\boldsymbol{R}\,\mathbf{t}} \coloneqq B_{\text{hom}}(S) \, \boldsymbol{Q}_{\text{RHS}} \, X = \begin{bmatrix} B(S)\,\boldsymbol{Q}_{\text{RHS}}\,X \\ p\|\mathbf{c}\| \end{bmatrix}, \tag{34}$$

where using the decomposition of $\mathbf{y}$ (18), we note that

$$\boldsymbol{R}_O^\top \, \boldsymbol{M}_n \, \mathbf{y}_{\boldsymbol{R}\,\mathbf{t}} = p\,\|\mathbf{c}\| \begin{bmatrix} \boldsymbol{R}\,\mathbf{x}+\mathbf{t} \\ \alpha\,(r^2 - \|\mathbf{c}\|^2 - \|\mathbf{x}+\mathbf{t}\|^2) \\ 1 \end{bmatrix}, \tag{35}$$

where $\|\mathbf{x}+\mathbf{t}\|^2 = \|\boldsymbol{R}\,\mathbf{x}+\mathbf{t}\|^2$.

Keeping the extended $(n+2) \times (n+2)$ versions of $\boldsymbol{M}_n \in \mathrm{O}(n+1)$, $\boldsymbol{R}_O \in \mathrm{O}(n)$, $V \in G < \mathrm{O}(n+1)$, using the translation representation $\boldsymbol{T}$ (28) derived earlier, as well as $\mathbf{y}_{\boldsymbol{R}\,\mathbf{t}}$, and following the same order of operations on the LHS as on the RHS of (32) (first an $\mathrm{O}(n)$ action, and then translation), we find that

$$\boldsymbol{T} \, V \, B_{\text{hom}}(S) \, X = B_{\text{hom}}(S) \, \boldsymbol{Q}_{\text{RHS}} \, X. \quad \square \tag{36}$$

Thus, we have proved (32) for

$$\boldsymbol{Q}(\boldsymbol{R}, \mathbf{t}) \coloneqq \boldsymbol{T}(\mathbf{t})\,V(\boldsymbol{R}) = \boldsymbol{M}_n^\top \, \boldsymbol{R}_O \begin{bmatrix} \boldsymbol{R} & 0 & \mathbf{t} \\ -2\alpha\,\mathbf{t}^\top\,\boldsymbol{R} & 1 & -\alpha\|\mathbf{t}\|^2 \\ \mathbf{0}^\top & 0 & 1 \end{bmatrix} \boldsymbol{R}_O^\top \, \boldsymbol{M}_n, \tag{37}$$

where $\boldsymbol{M}_n$, $\boldsymbol{R}_O$, and $V$ (16) are extended to $(n+2) \times (n+2)$ using the identity complement, $\boldsymbol{T}$ is given by (28), and $\alpha = \frac{\sqrt{n+1}}{2p\,\|\mathbf{c}\|} = \frac{\sqrt{n}}{2\|\mathbf{c}\|}$.

Similarly to what we did with $\boldsymbol{T}(\mathbf{t})$ in Section 3.2, we verify that $\boldsymbol{Q}(\boldsymbol{R}, \mathbf{t})$ is a linear representation by (a) finding its inverse

$$\boldsymbol{Q}(\boldsymbol{R}, \mathbf{t})^{-1} = \boldsymbol{M}_n^\top \, \boldsymbol{R}_O \begin{bmatrix} \boldsymbol{R}^\top & 0 & -\boldsymbol{R}^\top\,\mathbf{t} \\ 2\alpha\,\mathbf{t}^\top & 1 & -\alpha\|\mathbf{t}\|^2 \\ \mathbf{0}^\top & 0 & 1 \end{bmatrix} \boldsymbol{R}_O^\top \, \boldsymbol{M}_n, \tag{38}$$

which is valid for all $\boldsymbol{R}$ $in$ $\mathrm{O}(n)$, $\mathbf{t} \in \mathbb{R}^n$ and $\alpha$ with $\|\mathbf{c}\| \neq 0$ (which is the degenerate case considered in Section 3.2), and (b) confirming that the action of $\boldsymbol{Q}(\boldsymbol{R}_2, \mathbf{t}_2)\,\boldsymbol{Q}(\boldsymbol{R}_1, \mathbf{t}_1)$ results in the transformation of the original $n\mathrm{D}$ $\mathbf{x}$ as $\boldsymbol{R}_2(\boldsymbol{R}_1\,\mathbf{x}+\mathbf{t}_1)+\mathbf{t}_2 = \boldsymbol{R}_2\,\boldsymbol{R}_1\,\mathbf{x}+\boldsymbol{R}_2\,\mathbf{t}_1+\mathbf{t}_2$, i.e.,

$$\boldsymbol{Q}(\boldsymbol{R}_2, \mathbf{t}_2)\,\boldsymbol{Q}(\boldsymbol{R}_1, \mathbf{t}_1) = \boldsymbol{Q}(\boldsymbol{R}_2\,\boldsymbol{R}_1, \, \boldsymbol{R}_2\,\mathbf{t}_1+\mathbf{t}_2). \quad \square \tag{39}$$

This completes a constructive existence proof of an explicit $\mathrm{E}(n)$-equivariant representation for spherical decision surfaces, without relying on input centring or pairwise coordinate differences. For geometric interpretation, see Figure 1.

### 3.4 INVARIANT OPERATORS

For the $\mathrm{E}(n)$-equivariant feature in our case (25), we can no longer take the inner product of the input with itself to obtain an invariant feature as it is possible with $\mathrm{O}(n)$-equivariant features (as performed in Melnyk et al. (2024a;b)), and thus require other invariant operators.

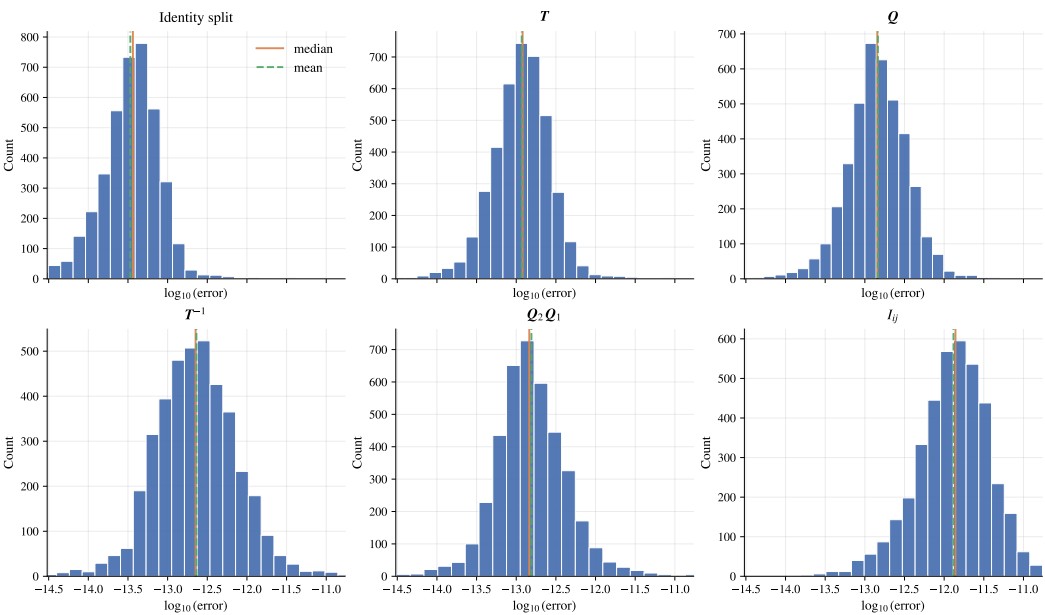

Figure 2: Numerical verification of our construction with histograms of $\log_{10}$ residual norms. Left-to-right and row-by-row, we report errors for: the invariant/equivariant split (identity split) (18), translation equivariance (27) with $T$, the equivariance constraint (32) with full $\mathrm{E}(n)$ actions $Q$, the inverse of the translation representation $T^{-1}$ (30), group composition $Q_2 Q_1$ (39), and invariants $I_{ij,k}$ (42) and (43). Residuals cluster at machine precision, confirming the derived identities up to numerical roundoff.

**Canonical triplets and translation law**   Within the neuron-specific (indexed with $k$) canonical frame, i.e., after the change of basis $R_O^{k\top} M_n$ (34)→(35) and normalisation by $p\|\mathbf{c}_k\|$, the homogeneous feature for an input point $\mathbf{x}_i \in \mathbb{R}^n$ splits into a vector–scalar triple $[\,\mathbf{x}_i;\; s_{ik};\; 1\,]$. Under a global translation $\mathbf{x}_i \mapsto \mathbf{x}_i + \mathbf{t}$, the equivariant representation acts as

$$\mathbf{x}_i \mapsto \mathbf{x}_i + \mathbf{t}, \qquad s_{ik} \mapsto s_{ik} - 2\alpha_k\, \mathbf{t}^\top \mathbf{x}_i - \alpha_k \|\mathbf{t}\|^2, \tag{40}$$

where $\alpha_k = \frac{\sqrt{n}}{2\|\mathbf{c}_k\|}$ and $s_{ik} = \alpha_k(r_k^2 - \|\mathbf{c}_k\|^2 - \|\mathbf{x}_i\|^2)$.

**Single-point invariant**   A per-neuron scalar

$$I_{i,k} := s_{ik} + \alpha_k\|\mathbf{x}_i\|^2 = \alpha_k\big(r_k^2 - \|\mathbf{c}_k\|^2\big) \tag{41}$$

thus remains invariant under translations and rotations/reflections. It serves as a learned bias/radius term for normalisation or gating across neurons.

**Pairwise invariant and distance form**   For two points $i, j$ under the same neuron $k$, we define

$$I_{ij,k} := s_{ik} + s_{jk} + 2\alpha_k\, \mathbf{x}_i^\top \mathbf{x}_j, \tag{42}$$

which is exactly translation-invariant; expanding $s_{ik}$ yields the rotation/reflection-invariant distance form

$$I_{ij,k} = \alpha_k\Big(2r_k^2 - 2\|\mathbf{c}_k\|^2 - \|\mathbf{x}_i - \mathbf{x}_j\|^2\Big), \tag{43}$$

so $I_{ij,k}$ depends only on the pairwise distance $\|\mathbf{x}_i - \mathbf{x}_j\|$ and neuron parameters $\mathbf{c}_k$ and $r_k$.

### 3.5   STRUCTURAL IMPLICATIONS

The results of this work are best interpreted at the level of individual geometric constructions. In particular, enforcing exact $\mathrm{E}(n)$-equivariance for non-linear spherical decision surfaces leads to representations $Q(R, \mathbf{t})$ (37) that depend explicitly on the learned geometric parameters $\mathbf{c}_k \in \mathbb{R}^n$ of each neuron. This dependence is intrinsic to the construction and reflects the fact that the geometry of the decision surface itself is learned. As a consequence, the output vector $\begin{bmatrix} \mathbf{y} \\ p\|\mathbf{c}\| \end{bmatrix} \in \mathbb{R}^{n+2}$ (25) is

neither in the same space as the input $\mathbf{x} \in \mathbb{R}^n$ nor in the conformal space in which the embedding $\boldsymbol{X} \in \mathbb{R}^{n+2}$ (7) resides (since the last two parameters of the output are not as they should be in the embedding). For this reason, $B_{\text{hom}}(\widetilde{\boldsymbol{S}})$ cannot be interpreted as a drop-in neural network layer whose outputs can be directly stacked or composed in the usual way. We therefore view $B_{\text{hom}}(\widetilde{\boldsymbol{S}})$ primarily as a theoretically well-defined geometric primitive rather than a drop-in neural network layer. Possible ways of integrating such primitives into larger equivariant systems (e.g., via canonicalisation and invariant extraction) can be envisaged but are not pursued further in this work.

## 4   DEMONSTRATION

**Numerical verification**    As a sanity check of the derived operators, we use PyTorch (Paszke et al., 2019) and numerically validate the identities from Section 3 by sampling random $\mathbf{x} \in \mathbb{R}^n$ (where required, $\mathbf{x}_1$ and $\mathbf{x}_2$), $\boldsymbol{R} \in \mathrm{O}(n)$, $\mathbf{t} \in \mathbb{R}^n$, $\mathbf{c} \in \mathbb{R}^n$, and $r \in \mathbb{R}$, and reporting the distribution of $\log_{10}$ residual norms for each check. Each check is performed 1000 times for each dimension $n = \{2, 3, 4, 5\}$. The resulting histograms presented in (Figure 2) show that residuals concentrate at machine precision, supporting the correctness of our derived operators.

**3D Tetris data classification**    We then perform a minimal proof-of-concept experiment verifying that the derived $\mathrm{E}(n)$-invariant primitives (42) behave as predicted in practice. We solve a classification task on a modified version of the 3D Tetris dataset used by Thomas et al. (2018): we remove the mirrored chiral copy to avoid reflection-sensitivity since our demonstration targets $\mathrm{E}(n)$-invariance. Each sample is a 4-point shape in $\mathbb{R}^3$, and 7 distinct shapes in total constitute the entire training set. The test set consists of the Tetris shapes subjected to random rotations/reflections and translations, with 7000 samples in total.

We build our model using one $\mathrm{E}(3)$-invariant bank with $K = 10$ neurons (26) (with randomly initialised $\boldsymbol{S}_k$ and each $\gamma_k$ initially set to 1), producing closed-form invariants (42), followed by a single linear head classifying from these invariants, in total containing 127 parameters. As a baseline, we construct a 3-layer multilayer perceptron (MLP) with ReLU activations and the same head size, yielding a 191-parameter network. We train three versions of this MLP: (i) as-is (MLP), (ii) with on-the-fly $\mathrm{E}(3)$-augmentation (MLP+aug), and (iii) with centring the input and using $\mathrm{O}(3)$-augmentation (MLP+centr+aug). All the models follow the per-point DeepSet (Zaheer et al., 2017) construction, and are therefore permutation-invariant (we use mean pooling). We train the models for 1000 epochs by minimising the cross-entropy and using the Adam optimiser with the default hyperparameters. The results in Table 1 show that the derived invariant primitives behave as theoretically expected under $\mathrm{E}(n)$ transformations, while standard MLPs rely on additional mechanisms to achieve invariance.

Table 1: Downstream classification accuracy on the 3D Tetris dataset under unseen $\mathrm{E}(n)$ transformations. Chance level is $1/7 \approx 14.3\%$.

| Model | Test Accuracy (%) |
|---|---|
| MLP | 14.7 |
| MLP+aug | 15.9 |
| MLP+centr+aug | 36.5 |
| **Ours** | **100.0** |

## 5   CONCLUSION

In this work, we presented a constructive derivation of exactly $\mathrm{E}(n)$-equivariant spherical decision surfaces, focusing on explicit representations and geometric correctness rather than architectural design. We hope this work serves as a theoretical reference point for future investigations into geometric decision surfaces and equivariant representations.

## ACKNOWLEDGEMENTS

This work was supported by the Wallenberg AI, Autonomous Systems and Software Program (WASP) and the Wallenberg Initiative Materials Science for Sustainability (WISE), by the Swedish Research Council through a grant for the project Uncertainty-Aware Transformers for Regression Tasks in Computer Vision (2022-04266), and the strategic research environment ELLIIT. In addition, we thank the reviewers for their helpful feedback.

IMPACT STATEMENT

This paper presents work aiming to advance the field of machine learning. There are many potential societal consequences of our work, none of which we feel must be specifically highlighted here, perhaps apart from materials science applications, where the development of new materials might have a significant impact on sustainability.

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

## A COMPLETE DERIVATIONS

In this section, we present complete derivation and proofs, for convenience, following the same organisation as Section 3 and restating expressions.

### A.1 IDENTITY SPLIT: EQUIVARIANT HYPERSPHERE OUTPUT DECOMPOSITION INTO IRREPS

Given $\boldsymbol{X} \in \mathbb{R}^{n+2}$ (7) and $\mathbf{y} := B(\boldsymbol{S})\,\boldsymbol{X} \in \mathbb{R}^{n+1}$, we decompose it into $\mathrm{O}(n)$-equivariant and invariant parts using (14) as

$$
\boldsymbol{R}_O^\top \boldsymbol{M}_n \, \mathbf{y} = \boldsymbol{R}_O^\top \boldsymbol{M}_n \, B(\boldsymbol{S}) \, \boldsymbol{X} = \boldsymbol{R}_O^\top \boldsymbol{M}_n \left[ \begin{bmatrix} \|\mathbf{c}\| \, \boldsymbol{P}_n^\top & s_{n+1} \, \mathbf{1} \end{bmatrix} \boldsymbol{R}_O & \mathbf{1} \right] \boldsymbol{X}
$$

$$
= \boldsymbol{R}_O^\top \left[ \begin{bmatrix} p\,\|\mathbf{c}\| \begin{bmatrix} \mathbf{I}_n \\ \mathbf{0}^\top \end{bmatrix} & \mathbf{0} \\ & p\sqrt{n}\,s_{n+1} \end{bmatrix} \boldsymbol{R}_O & \mathbf{0} \\ & p\sqrt{n} \end{bmatrix} \boldsymbol{X}
$$

$$
= \begin{bmatrix} \begin{bmatrix} p\,\|\mathbf{c}\| \, \mathbf{I}_n & \mathbf{0} \\ \mathbf{0}^\top & p\sqrt{n}\,s_{n+1} \end{bmatrix} \boldsymbol{R}_O^\top \boldsymbol{R}_O & \mathbf{0} \\ & p\sqrt{n} \end{bmatrix} \boldsymbol{X} = \begin{bmatrix} p\,\|\mathbf{c}\| \, \mathbf{I}_n & \mathbf{0} & \mathbf{0} \\ \mathbf{0}^\top & p\sqrt{n}\,s_{n+1} & p\sqrt{n} \end{bmatrix} \boldsymbol{X} \quad (44)
$$

$$
= \begin{bmatrix} p\,\|\mathbf{c}\| \, \mathbf{I}_n & \mathbf{0} & \mathbf{0} \\ \mathbf{0}^\top & p\sqrt{n}\,\tfrac{1}{2}(\|\mathbf{c}\|^2 - r^2) & p\sqrt{n} \end{bmatrix} \boldsymbol{X} = \begin{bmatrix} p\,\|\mathbf{c}\| \, \mathbf{x} \\ \tfrac{p\sqrt{n}}{2}(r^2 - \|\mathbf{c}\|^2 - \|\mathbf{x}\|^2) \end{bmatrix}
$$

$$
= p\,\|\mathbf{c}\| \begin{bmatrix} \mathbf{x} \\ \tfrac{\sqrt{n}}{2\|\mathbf{c}\|}(r^2 - \|\mathbf{c}\|^2 - \|\mathbf{x}\|^2) \end{bmatrix} = p\,\|\mathbf{c}\| \begin{bmatrix} \mathbf{x} \\ \alpha\,(r^2 - \|\mathbf{c}\|^2 - \|\mathbf{x}\|^2) \end{bmatrix},
$$

where $\alpha := \frac{\sqrt{n+1}}{2p\,\|\mathbf{c}\|} = \frac{\sqrt{n}}{2\,\|\mathbf{c}\|}$, and the invariant part is $\frac{\sqrt{n+1}}{2}(r^2 - \|\mathbf{c}\|^2 - \|\mathbf{x}\|^2)$, with $\mathbf{c}$ and $r$ being learnable parameters.

Observe that in the case $\|\mathbf{c}\| = 0$, the $\mathrm{O}(n)$-equivariant part, i.e., the original vector $\mathbf{x}$, zeroes out, and what we are left with is

$$
\boldsymbol{R}_O^\top \boldsymbol{M}_n \, \mathbf{y} = \boldsymbol{R}_O^\top \boldsymbol{M}_n \, B(\boldsymbol{S})\,\boldsymbol{X} = \begin{bmatrix} p\,0\,\mathbf{x} \\ \tfrac{p\sqrt{n}}{2}(r^2 - 0 - \|\mathbf{x}\|^2) \end{bmatrix} = \begin{bmatrix} \mathbf{0} \\ \tfrac{p\sqrt{n}}{2}(r^2 - \|\mathbf{x}\|^2) \end{bmatrix} . \quad (45)
$$

Also, if $\boldsymbol{X}$ is transformed with $\boldsymbol{R} \in \mathrm{O}(n)$ extended to $(n+2) \times (n+2)$ using the identity matrix, as $\boldsymbol{R}\,\boldsymbol{X}$, defining $\mathbf{y}_{\boldsymbol{R}} := B(\boldsymbol{S})\,\boldsymbol{R}\,\boldsymbol{X}$, we obtain

$$
\boldsymbol{R}_O^\top \boldsymbol{M}_n \, \mathbf{y}_{\boldsymbol{R}} = \boldsymbol{R}_O^\top \boldsymbol{M}_n \, B(\boldsymbol{S})\,\boldsymbol{R}\,\boldsymbol{X} = p\,\|\mathbf{c}\| \begin{bmatrix} \boldsymbol{R}\,\mathbf{x} \\ \alpha\,(r^2 - \|\mathbf{c}\|^2 - \|\mathbf{x}\|^2) \end{bmatrix} . \quad (46)
$$

### A.2 TRANSLATION EQUIVARIANCE

The desired equivariance property is

$$
\boldsymbol{T}(\mathbf{t})\,B(\boldsymbol{S})\,\boldsymbol{X} = B(\boldsymbol{S})\,\boldsymbol{T}_{\text{RHS}}\,\boldsymbol{X}, \quad (47)
$$

where $\boldsymbol{T}(\mathbf{t})$ is the translation representation we seek and

$$
\boldsymbol{T}_{\text{RHS}} = \begin{bmatrix} \mathbf{I}_n & -\mathbf{t} & \mathbf{0} \\ \mathbf{0}^\top & 1 & 0 \\ -\mathbf{t}^\top & \tfrac{1}{2}\|\mathbf{t}\|^2 & 1 \end{bmatrix} \quad (48)
$$

is the $(n+2) \times (n+2)$ matrix (9) with $\boldsymbol{R} = \mathbf{I}_n$.

Consider the RHS and denote $\mathbf{y_t}$ to be the output of the equivariant hypersphere given the input $\mathbf{x}$ translated as $\mathbf{x} + \mathbf{t}$ and embedded in $\mathbb{R}^{n+2}$ (see (7)) as $\boldsymbol{X_t}$:

$$
\mathbf{y_t} := B(\boldsymbol{S})\,\boldsymbol{X}_t = B(\boldsymbol{S})\,\boldsymbol{T}_{\text{RHS}}(\mathbf{t})\,\boldsymbol{X} \in \mathbb{R}^{n+1}. \quad (49)
$$

In the decomposition of $\mathbf{y}$ (44), we can left-multiply both sides with some $(n+1) \times (n+1)$ matrix $\boldsymbol{T}_{\text{interm}}$:

$$\boldsymbol{T}_{\text{interm}} \ \boldsymbol{R}_O^T \ \boldsymbol{M} \ \mathbf{y} = p \, \|\mathbf{c}\| \ \boldsymbol{T}_{\text{interm}} \begin{bmatrix} \mathbf{x} \\ \alpha \, (r^2 - \|\mathbf{c}\|^2 - \|\mathbf{x}\|^2) \end{bmatrix}. \tag{50}$$

From this decomposition, it is clear that in order to act as a translation, $\mathbf{t}$, on the original $\mathbf{x}$, $\boldsymbol{T}_{\text{interm}}$ must transform the expression in the following manner:

$$\begin{aligned} p \, \|\mathbf{c}\| \ \boldsymbol{T}_{\text{interm}} \begin{bmatrix} \mathbf{x} \\ \alpha \, (r^2 - \|\mathbf{c}\|^2 - \|\mathbf{x}\|^2) \end{bmatrix} &= p \, \|\mathbf{c}\| \begin{bmatrix} \mathbf{x} + \mathbf{t} \\ \alpha \, (r^2 - \|\mathbf{c}\|^2 - \|\mathbf{x} + \mathbf{t}\|^2) \end{bmatrix} \\ &= \boldsymbol{R}_O^\top \, \boldsymbol{M}_n \, \mathbf{y_t} = \boldsymbol{R}_O^\top \, \boldsymbol{M}_n \, B(\boldsymbol{S}) \, \boldsymbol{T}_{\text{RHS}} \, \boldsymbol{X} \,. \end{aligned} \tag{51}$$

It is achievable with an *affine* transformation, i.e., if instead of left multiplying with $\boldsymbol{T}_{\text{interm}}$, we added $p \, \|\mathbf{c}\| \begin{bmatrix} \mathbf{t} \\ -2\alpha \, \mathbf{t}^\top \, \mathbf{x} - \alpha \|\mathbf{t}\|^2 \end{bmatrix}$ to the decomposition. This would, however, need to depend on the data vector $\mathbf{x}$. However, this is *impossible* with a linear transformation, that is, in the original $(n+1)$D space, which means there is not such a translation representation $\boldsymbol{T}(\mathbf{t}) \in \mathbb{R}^{(n+1) \times (n+1)}$ to satisfy (47).

**Homogeneous representation to the rescue**    If in (51) we instead use a homogeneous representation

$$p \, \|\mathbf{c}\| \begin{bmatrix} \mathbf{x} \\ \alpha \, (r^2 - \|\mathbf{c}\|^2 - \|\mathbf{x}\|^2) \\ 1 \end{bmatrix}, \tag{52}$$

resulting in

$$\begin{aligned} p \, \|\mathbf{c}\| \ \boldsymbol{T}_{\text{interm}} \begin{bmatrix} \mathbf{x} \\ \alpha \, (r^2 - \|\mathbf{c}\|^2 - \|\mathbf{x}\|^2) \\ 1 \end{bmatrix} &= p \, \|\mathbf{c}\| \begin{bmatrix} \mathbf{x} + \mathbf{t} \\ \alpha \, (r^2 - \|\mathbf{c}\|^2 - \|\mathbf{x} + \mathbf{t}\|^2) \\ 1 \end{bmatrix} \\ &= p \, \|\mathbf{c}\| \begin{bmatrix} \mathbf{x} + \mathbf{t} \\ \alpha \, (r^2 - \|\mathbf{c}\|^2 - \|\mathbf{x}\|^2 - 2\|\mathbf{x}\|^\top \, \mathbf{t} - \|\mathbf{t}\|^2) \\ 1 \end{bmatrix}, \end{aligned} \tag{53}$$

we can find a linear transformation $\boldsymbol{T}_{\text{interm}}$ that satisfies this equation.

Solving for each row of $\boldsymbol{T}_{\text{interm}}$ in (53), we obtain the $(n+2) \times (n+2)$

$$\boldsymbol{T}_{\text{interm}}(\mathbf{t}) = \begin{bmatrix} \mathbf{I}_n & 0 & \mathbf{t} \\ -2\alpha \, \mathbf{t}^\top & 1 & -\alpha \|\mathbf{t}\|^2 \\ \mathbf{0}^\top & 0 & 1 \end{bmatrix}, \tag{54}$$

where $\alpha = \frac{\sqrt{n+1}}{2p \, \|\mathbf{c}\|} = \frac{\sqrt{n}}{2 \, \|\mathbf{c}\|}$.

Subsequently, we can write the decomposition (50) backwards as

$$\begin{aligned} p \, \|\mathbf{c}\| \ \boldsymbol{T}_{\text{interm}} \begin{bmatrix} \mathbf{x} \\ \alpha \, (r^2 - \|\mathbf{c}\|^2 - \|\mathbf{x}\|^2) \\ 1 \end{bmatrix} &= p \, \|\mathbf{c}\| \ \boldsymbol{T}_{\text{interm}} \begin{bmatrix} \boldsymbol{R}_O^\top \boldsymbol{M} & \mathbf{0} \\ \mathbf{0}^\top & 1 \end{bmatrix} \begin{bmatrix} \frac{\mathbf{y}}{p \, \|\mathbf{c}\|} \\ 1 \end{bmatrix} \\ &= \boldsymbol{T}_{\text{interm}} \begin{bmatrix} \boldsymbol{R}_O^\top \boldsymbol{M}_n & \mathbf{0} \\ \mathbf{0}^\top & 1 \end{bmatrix} \begin{bmatrix} B(\boldsymbol{S}) \, \boldsymbol{X} \\ p \, \|\mathbf{c}\| \end{bmatrix}. \end{aligned} \tag{55}$$

To obtain the homogeneous representation of $\mathbf{y}$ in the equation above (55), $\begin{bmatrix} \mathbf{y} \\ p\,\|\mathbf{c}\| \end{bmatrix}$, we need to modify the original $(n+1)\times(n+2)$ equivariant hypersphere $B(\boldsymbol{S})$ (12), and create the $(n+2)\times(n+2)$ $B_{\text{hom}}(\boldsymbol{S})$:

$$B_{\text{hom}}(\boldsymbol{S}) := \begin{bmatrix} & B(\boldsymbol{S}) & \\ \mathbf{0}^\top & -p\,\|\mathbf{c}\| & 0 \end{bmatrix}, \tag{56}$$

so that

$$B_{\text{hom}}(\boldsymbol{S})\,X = \begin{bmatrix} B(\boldsymbol{S})\,X \\ p\,\|\mathbf{c}\| \end{bmatrix} = \begin{bmatrix} \mathbf{y} \\ p\,\|\mathbf{c}\| \end{bmatrix} \in \mathbb{R}^{n+2}, \tag{57}$$

where $\mathbf{c} \in \mathbb{R}^n$ are parameters in $\boldsymbol{S}$, and hence no new learnable parameters are added, and the $-$ sign in the $-p\,\|\mathbf{c}\|$ element in $B_{\text{hom}}(\boldsymbol{S})$ is due to the second last element of the embedding $X$ (7) being $-1$.

Observe that with respect to the input vector $\mathbf{x} \in \mathbb{R}^n$, the equivariant hypersphere (56) $\mathbf{F}_n(\,\cdot\,;\boldsymbol{S})$ : $\mathbb{R}^{n+2} \to \mathbb{R}^{n+2}$ represents a non-linear E$(n)$-equivariant function. In the general case with the learnable $\gamma \in \mathbb{R}$ parameter (see Section 2.3), we define $B_{\text{hom}}(\widetilde{\boldsymbol{S}})$ as

$$B_{\text{hom}}(\widetilde{\boldsymbol{S}}) := \gamma \begin{bmatrix} & B(\boldsymbol{S}) & \\ \mathbf{0}^\top & -p\,\|\mathbf{c}\| & 0 \end{bmatrix}. \tag{58}$$

We rewrite the desired equivariance property (47) for the new $B_{\text{hom}}(\boldsymbol{S})$:

$$\boldsymbol{T}(\mathbf{t})B_{\text{hom}}(\boldsymbol{S})\,X = B_{\text{hom}}(\boldsymbol{S})\,\boldsymbol{T}_{\text{RHS}}\,X. \tag{59}$$

We first start with the LHS: To complete the $(n+2) \times (n+2)$ representation $\boldsymbol{T}(\mathbf{t})$ on the LHS of (59), we complete the last expression of (55), i.e., rotate the $(n+1)$D part back to the original coordinate system of $\mathbf{y}$ using $\boldsymbol{M}_n^\top \boldsymbol{R}_O$:

$$
\begin{aligned}
\boldsymbol{T}(\mathbf{t})B_{\text{hom}}(\boldsymbol{S})\,X &= \begin{bmatrix} \boldsymbol{M}_n^\top \boldsymbol{R}_O & \mathbf{0} \\ \mathbf{0}^\top & 1 \end{bmatrix} \boldsymbol{T}_{\text{interm}} \begin{bmatrix} \boldsymbol{R}_O^\top \boldsymbol{M}_n & \mathbf{0} \\ \mathbf{0}^\top & 1 \end{bmatrix} B_{\text{hom}}(\boldsymbol{S})\,X \\
&= p\,\|\mathbf{c}\| \begin{bmatrix} \boldsymbol{M}_n^\top \boldsymbol{R}_O & \mathbf{0} \\ \mathbf{0}^\top & 1 \end{bmatrix} \begin{bmatrix} \mathbf{I}_n & 0 & \mathbf{t} \\ -2\alpha\,\mathbf{t}^\top & 1 & -\alpha\|\mathbf{t}\|^2 \\ \mathbf{0}^\top & 0 & 1 \end{bmatrix} \begin{bmatrix} \mathbf{x} \\ \alpha\,(r^2 - \|\mathbf{c}\|^2 - \|\mathbf{x}\|^2) \\ 1 \end{bmatrix} \\
&= p\,\|\mathbf{c}\| \begin{bmatrix} \boldsymbol{M}_n^\top \boldsymbol{R}_O & \mathbf{0} \\ \mathbf{0}^\top & 1 \end{bmatrix} \begin{bmatrix} \mathbf{x}+\mathbf{t} \\ \alpha\,(r^2 - \|\mathbf{c}\|^2 - \|\mathbf{x}+\mathbf{t}\|^2) \\ 1 \end{bmatrix} \\
&= \begin{bmatrix} \boldsymbol{M}_n^\top \boldsymbol{R}_O & \mathbf{0} \\ \mathbf{0}^\top & 1 \end{bmatrix} \begin{bmatrix} \boldsymbol{R}_O^\top \boldsymbol{M}_n\,\mathbf{y_t} \\ p\,\|\mathbf{c}\| \end{bmatrix} = \begin{bmatrix} \boldsymbol{M}_n^\top \boldsymbol{R}_O & \mathbf{0} \\ \mathbf{0}^\top & 1 \end{bmatrix} \begin{bmatrix} \boldsymbol{R}_O^\top \boldsymbol{M}_n & \mathbf{0} \\ \mathbf{0}^\top & 1 \end{bmatrix} \begin{bmatrix} \mathbf{y_t} \\ p\,\|\mathbf{c}\| \end{bmatrix} \\
&= \mathbf{I}_{n+2} \begin{bmatrix} B(\boldsymbol{S})\,\boldsymbol{T}_{\text{RHS}}\,X \\ p\,\|\mathbf{c}\| \end{bmatrix} = B_{\text{hom}}(\boldsymbol{S})\,\boldsymbol{T}_{\text{RHS}}\,X, \qquad\qquad \square
\end{aligned}
\tag{60}
$$

where $\mathbf{y_t} \in \mathbb{R}^{n+1}$ is defined in (49), effectively proving (59) for $B_{\text{hom}}(\boldsymbol{S})$ (56) and

$$\boldsymbol{T}(\mathbf{t}) := \boldsymbol{M}_n^\top\,\boldsymbol{R}_O\,\boldsymbol{T}_{\text{interm}}\,\boldsymbol{R}_O^\top\,\boldsymbol{M}_n, \tag{61}$$

where $\boldsymbol{T}_{\text{interm}}$ is given by (54) and, gently abusing notation, $\boldsymbol{M}_n \in \text{O}(n+1)$ and $\boldsymbol{R}_O \in \text{O}(n)$ are extended to $(n+2) \times (n+2)$ using the identity matrix.

We verify that $\boldsymbol{T}(\mathbf{t})$ is indeed a linear representation as necessary for the equivariance definition (5), by (a) finding its inverse

$$
\begin{aligned}
\boldsymbol{T}(\mathbf{t})^{-1} &= \boldsymbol{M}_n^\top\,\boldsymbol{R}_O\,\boldsymbol{T}_{\text{interm}}^{-1}\,\boldsymbol{R}_O^\top\,\boldsymbol{M}_n, \\
\boldsymbol{T}_{\text{interm}}^{-1} &= \begin{bmatrix} \mathbf{I}_n & 0 & -\mathbf{t} \\ 2\alpha\,\mathbf{t}^\top & 1 & -\alpha\|\mathbf{t}\|^2 \\ \mathbf{0}^\top & 0 & 1 \end{bmatrix},
\end{aligned}
\tag{62}
$$

which is valid for all $\mathbf{t} \in \mathbb{R}^n$ and $\alpha$ with $\|\mathbf{c}\| \neq 0$, and (b) verifying that the action of $\boldsymbol{T}(\mathbf{t}_2)\,\boldsymbol{T}(\mathbf{t}_1)$ results in the translation of the original $n$D $\mathbf{x}$ as $\mathbf{x} + \mathbf{t}_1 + \mathbf{t}_2$:

$$
\begin{aligned}
\boldsymbol{T}(\mathbf{t}_2)\,\boldsymbol{T}(\mathbf{t}_1) &= \boldsymbol{M}_n^\top\,\boldsymbol{R}_O\,\boldsymbol{T}_{\text{interm}}(\mathbf{t}_2)\,\boldsymbol{R}_O^\top\,\boldsymbol{M}_n\,\boldsymbol{M}_n^\top\,\boldsymbol{R}_O\,\boldsymbol{T}_{\text{interm}}(\mathbf{t}_1)\,\boldsymbol{R}_O^\top\,\boldsymbol{M}_n \\
&= \boldsymbol{M}_n^\top\,\boldsymbol{R}_O\,\boldsymbol{T}_{\text{interm}}(\mathbf{t}_2)\,\boldsymbol{T}_{\text{interm}}(\mathbf{t}_1)\,\boldsymbol{R}_O^\top\,\boldsymbol{M}_n \\
&= \boldsymbol{M}_n^\top\,\boldsymbol{R}_O \begin{bmatrix} \mathbf{I}_n & 0 & \mathbf{t}_2 \\ -2\alpha\,\mathbf{t}_2^\top & 1 & -\alpha\|\mathbf{t}_2\|^2 \\ \mathbf{0}^\top & 0 & 1 \end{bmatrix} \begin{bmatrix} \mathbf{I}_n & 0 & \mathbf{t}_1 \\ -2\alpha\,\mathbf{t}_1^\top & 1 & -\alpha\|\mathbf{t}_1\|^2 \\ \mathbf{0}^\top & 0 & 1 \end{bmatrix} \boldsymbol{R}_O^\top\,\boldsymbol{M}_n \\
&= \boldsymbol{M}_n^\top\,\boldsymbol{R}_O \begin{bmatrix} \mathbf{I}_n & 0 & \mathbf{t}_1 + \mathbf{t}_2 \\ -2\alpha(\mathbf{t}_1+\mathbf{t}_2)^\top & 1 & -\alpha\|\mathbf{t}_1+\mathbf{t}_2\|^2 \\ \mathbf{0}^\top & 0 & 1 \end{bmatrix} \boldsymbol{R}_O^\top\,\boldsymbol{M}_n \\
&= \boldsymbol{M}_n^\top\,\boldsymbol{R}_O\,\boldsymbol{T}_{\text{interm}}(\mathbf{t}_1+\mathbf{t}_2)\,\boldsymbol{R}_O^\top\,\boldsymbol{M}_n\ . \hspace{2cm} \square
\end{aligned}
\tag{63}
$$

### A.3    Completing to E($n$)-Equivariance

The desired equivariance property is

$$
\boldsymbol{Q}(\boldsymbol{R}, \mathbf{t})\,B_{\text{hom}}(\boldsymbol{S})\,\boldsymbol{X} = B_{\text{hom}}(\boldsymbol{S})\,\boldsymbol{Q}_{\text{RHS}}\,\boldsymbol{X},
\tag{64}
$$

where $\boldsymbol{Q}_{\text{RHS}} \coloneqq \boldsymbol{Q}_X$ (9).

By using an identity complement and extending $\boldsymbol{V}$ (16) to $(n+2) \times (n+2)$ on the LHS in (15), we note that $B_{\text{hom}}(\boldsymbol{S})$ (24) is also O($n$)-equivariant: since the RHS in (15) can be rewritten as

$$
B_{\text{hom}}(\boldsymbol{S})\,\boldsymbol{R}\,\boldsymbol{X} = \begin{bmatrix} & B(\boldsymbol{S}) & \\ \mathbf{0}^\top & -p\|\mathbf{c}\| & 0 \end{bmatrix} \begin{bmatrix} \boldsymbol{R}\,\mathbf{x} \\ -1 \\ -\frac{1}{2}\|\mathbf{x}\|^2 \end{bmatrix} = \begin{bmatrix} B(\boldsymbol{S})\,\boldsymbol{R}\,\boldsymbol{X} \\ p\|\mathbf{c}\| \end{bmatrix},
\tag{65}
$$

where $p\|\mathbf{c}\| = \sqrt{\frac{n+1}{n}}\,\|\mathbf{c}\|$ and $\|\mathbf{x}\|^2 = \|\boldsymbol{R}\,\mathbf{x}\|^2$ are O($n$)-invariant. The same is valid for the LHS, where the additional $(n+2)$-th element in the resulting vector is also $p\|\mathbf{c}\|$.

In turn, the RHS of (64) is

$$
\mathbf{y}_{\boldsymbol{R}\mathbf{t}} \coloneqq B_{\text{hom}}(\boldsymbol{S})\,\boldsymbol{Q}_{\text{RHS}}\,\boldsymbol{X} = \begin{bmatrix} & B(\boldsymbol{S}) & \\ \mathbf{0}^\top & -p\|c\| & 0 \end{bmatrix} \begin{bmatrix} \boldsymbol{R}\,\mathbf{x} + \mathbf{t} \\ -1 \\ -\frac{1}{2}\|\mathbf{x}+\mathbf{t}\|^2 \end{bmatrix} = \begin{bmatrix} B(\boldsymbol{S})\,\boldsymbol{Q}_{\text{RHS}}\,\boldsymbol{X} \\ p\|\mathbf{c}\| \end{bmatrix},
\tag{66}
$$

where using the decomposition of $\mathbf{y}$ (44), we note that

$$
\boldsymbol{R}_O^\top\,\boldsymbol{M}_n\,\mathbf{y}_{\boldsymbol{R}\mathbf{t}} = \boldsymbol{R}_O^\top\,\boldsymbol{M}_n\,B(\boldsymbol{S})\,\boldsymbol{Q}_{\text{RHS}}\,\boldsymbol{X} = p\,\|\mathbf{c}\| \begin{bmatrix} \boldsymbol{R}\,\mathbf{x}+\mathbf{t} \\ \alpha\,(r^2 - \|\mathbf{c}\|^2 - \|\mathbf{x}+\mathbf{t}\|^2) \end{bmatrix},
\tag{67}
$$

where $\|\mathbf{x}+\mathbf{t}\|^2 = \|\boldsymbol{R}\,\mathbf{x}+\mathbf{t}\|^2$.

Keeping the extended $(n+2) \times (n+2)$ versions of $\boldsymbol{M}_n \in$ O($n+1$), $\boldsymbol{R}_O \in$ O($n$), $\boldsymbol{V} \in G <$ O($n+1$), using the translation representation $\boldsymbol{T}$ (61) derived earlier, as well as $\mathbf{y}_{\boldsymbol{R}\mathbf{t}}$, and following the same order of operations on the LHS as on the RHS of (64) (first an O($n$) action, and then translation), we

rewrite (64) starting from the LHS as

$$
\begin{aligned}
\boldsymbol{T}\,\boldsymbol{V}\,B_{\mathrm{hom}}(\boldsymbol{S})\,\boldsymbol{X} &= \boldsymbol{M}_n^\top\,\boldsymbol{R}_O\,\boldsymbol{T}_{\mathrm{interm}}\,\boldsymbol{R}_O^\top\,\boldsymbol{M}\boldsymbol{M}_n^\top\,\boldsymbol{R}_O\,\boldsymbol{R}\,\boldsymbol{R}_O^\top\,\boldsymbol{M}\,B_{\mathrm{hom}}(\boldsymbol{S})\,\boldsymbol{X} \\
&= \boldsymbol{M}_n^\top\,\boldsymbol{R}_O\,\boldsymbol{T}_{\mathrm{interm}}\,\boldsymbol{R}\,\boldsymbol{R}_O^\top\,\boldsymbol{M}_n\,B_{\mathrm{hom}}(\boldsymbol{S})\,\boldsymbol{X} \\
&= \boldsymbol{M}_n^\top\,\boldsymbol{R}_O
\begin{bmatrix}
\boldsymbol{I}_n & 0 & \mathbf{t} \\
-2\alpha\,\mathbf{t}^\top & 1 & -\alpha\|\mathbf{t}\|^2 \\
\mathbf{0}^\top & 0 & 1
\end{bmatrix}
\begin{bmatrix}
\boldsymbol{R} & 0 & 0 \\
\mathbf{0}^\top & 1 & 0 \\
\mathbf{0}^\top & 0 & 1
\end{bmatrix}
\boldsymbol{R}_O^\top\,\boldsymbol{M}_n\,B_{\mathrm{hom}}(\boldsymbol{S})\,\boldsymbol{X} \\
&= \boldsymbol{M}_n^\top\,\boldsymbol{R}_O
\begin{bmatrix}
\boldsymbol{R} & 0 & \mathbf{t} \\
-2\alpha\,\mathbf{t}^\top\boldsymbol{R} & 1 & -\alpha\|\mathbf{t}\|^2 \\
\mathbf{0}^\top & 0 & 1
\end{bmatrix}
\boldsymbol{R}_O^\top\,\boldsymbol{M}_n\,B_{\mathrm{hom}}(\boldsymbol{S})\,\boldsymbol{X} \\
&= p\,\|\mathbf{c}\|\,\boldsymbol{M}_n^\top\,\boldsymbol{R}_O
\begin{bmatrix}
\boldsymbol{R} & 0 & \mathbf{t} \\
-2\alpha\,\mathbf{t}^\top\boldsymbol{R} & 1 & -\alpha\|\mathbf{t}\|^2 \\
\mathbf{0}^\top & 0 & 1
\end{bmatrix}
\begin{bmatrix}
\mathbf{x} \\
\alpha\,(r^2-\|\mathbf{c}\|^2-\|\mathbf{x}\|^2) \\
1
\end{bmatrix} \\
&= p\,\|\mathbf{c}\|\,\boldsymbol{M}_n^\top\,\boldsymbol{R}_O
\begin{bmatrix}
\boldsymbol{R}\,\mathbf{x}+\mathbf{t} \\
\alpha\,(r^2-\|\mathbf{c}\|^2-\|\mathbf{x}+\mathbf{t}\|^2) \\
1
\end{bmatrix}
= \boldsymbol{M}_n^\top\,\boldsymbol{R}_O\,\boldsymbol{R}_O^\top\,\boldsymbol{M}_n
\begin{bmatrix}
\mathbf{y}_{\boldsymbol{R}\,\mathbf{t}} \\
p\,\|\mathbf{c}\|
\end{bmatrix} \\
&= \boldsymbol{I}_{n+2}
\begin{bmatrix}
B(\boldsymbol{S})\,\boldsymbol{Q}_{\mathrm{RHS}}\,\boldsymbol{X} \\
p\,\|\mathbf{c}\|
\end{bmatrix}
= B_{\mathrm{hom}}(\boldsymbol{S})\,\boldsymbol{Q}_{\mathrm{RHS}}\,\boldsymbol{X}\,. \qquad\qquad \square
\end{aligned}
$$
(68)

Thus, we have proved (64) for

$$
\boldsymbol{Q}(\boldsymbol{R},\mathbf{t}) := \boldsymbol{T}(\mathbf{t})\,\boldsymbol{V}(\boldsymbol{R}) = \boldsymbol{M}_n^\top\,\boldsymbol{R}_O
\begin{bmatrix}
\boldsymbol{R} & 0 & \mathbf{t} \\
-2\alpha\,\mathbf{t}^\top\boldsymbol{R} & 1 & -\alpha\|\mathbf{t}\|^2 \\
\mathbf{0}^\top & 0 & 1
\end{bmatrix}
\boldsymbol{R}_O^\top\,\boldsymbol{M}_n,
\tag{69}
$$

where $\boldsymbol{M}_n$, $\boldsymbol{R}_O$, and $\boldsymbol{V}$ (16) are extended to $(n+2)\times(n+2)$ using the identity complement, $\boldsymbol{T}$ is given by (61), and $\alpha = \frac{\sqrt{n+1}}{2p\,\|\mathbf{c}\|} = \frac{\sqrt{n}}{2\|\mathbf{c}\|}$.

Similarly to what we did with $\boldsymbol{T}(\mathbf{t})$ in Section 3.2, we verify that $\boldsymbol{Q}(\boldsymbol{R},\mathbf{t})$ is a linear representation by (a) finding its inverse

$$
\begin{aligned}
\boldsymbol{Q}(\boldsymbol{R},\mathbf{t})^{-1} &= \boldsymbol{V}(\boldsymbol{R})^{-1}\,\boldsymbol{T}(\mathbf{t})^{-1} = \boldsymbol{M}_n^\top\,\boldsymbol{R}_O\,\boldsymbol{R}^\top\,\boldsymbol{T}_{\mathrm{interm}}^{-1}\,\boldsymbol{R}_O^\top\,\boldsymbol{M}_n \\
&= \boldsymbol{M}_n^\top\,\boldsymbol{R}_O
\begin{bmatrix}
\boldsymbol{R}^\top & 0 & -\boldsymbol{R}^\top\,\mathbf{t} \\
2\alpha\,\mathbf{t}^\top & 1 & -\alpha\|\mathbf{t}\|^2 \\
\mathbf{0}^\top & 0 & 1
\end{bmatrix}
\boldsymbol{R}_O^\top\,\boldsymbol{M}_n,
\end{aligned}
\tag{70}
$$

which is valid for all $\boldsymbol{R}$ $in$ $\mathrm{O}(n)$, $\mathbf{t}\in\mathbb{R}^n$ and $\alpha$ with $\|\mathbf{c}\|\neq 0$ (which is the degenerate case considered in Section 3.2), and (b) confirming that the action of $\boldsymbol{Q}(\boldsymbol{R}_2,\mathbf{t}_2)\,\boldsymbol{Q}(\boldsymbol{R}_1,\mathbf{t}_1)$ results in the transformation

of the original $n$D $\mathbf{x}$ as $\boldsymbol{R}_2(\boldsymbol{R}_1\,\mathbf{x}+\mathbf{t}_1)+\mathbf{t}_2 = \boldsymbol{R}_2\,\boldsymbol{R}_1\,\mathbf{x}+\boldsymbol{R}_2\,\mathbf{t}_1+\mathbf{t}_2$:

$$\boldsymbol{Q}(\boldsymbol{R}_2,\mathbf{t}_2)\,\boldsymbol{Q}(\boldsymbol{R}_1,\mathbf{t}_1)$$

$$= \boldsymbol{M}_n^\top\,\boldsymbol{R}_O \begin{bmatrix} \boldsymbol{R}_2 & 0 & \mathbf{t}_2 \\ -2\alpha\,\mathbf{t}_2^\top\,\boldsymbol{R}_2 & 1 & -\alpha\|\mathbf{t}_2\|^2 \\ \mathbf{0}^\top & 0 & 1 \end{bmatrix} \boldsymbol{R}_O^\top\,\boldsymbol{M}_n\,\boldsymbol{M}_n^\top\,\boldsymbol{R}_O \begin{bmatrix} \boldsymbol{R}_1 & 0 & \mathbf{t}_1 \\ -2\alpha\,\mathbf{t}_1^\top\,\boldsymbol{R}_1 & 1 & -\alpha\|\mathbf{t}_1\|^2 \\ \mathbf{0}^\top & 0 & 1 \end{bmatrix} \boldsymbol{R}_O^\top\,\boldsymbol{M}_n$$

$$= \boldsymbol{M}_n^\top\,\boldsymbol{R}_O \begin{bmatrix} \boldsymbol{R}_2\,\boldsymbol{R}_1 & 0 & \boldsymbol{R}_2\,\mathbf{t}_1+\mathbf{t}_2 \\ -2\alpha\,\mathbf{t}_2^\top\,\boldsymbol{R}_2\,\boldsymbol{R}_1 -2\alpha\,\mathbf{t}_1^\top\,\boldsymbol{R}_1 & 1 & -2\alpha\,\mathbf{t}_2^\top\,\boldsymbol{R}_2\,\mathbf{t}_1 -\alpha\|\mathbf{t}_1\|^2 - \alpha\|\mathbf{t}_2\|^2 \\ \mathbf{0}^\top & 0 & 1 \end{bmatrix} \boldsymbol{R}_O^\top\,\boldsymbol{M}_n$$

$$= \boldsymbol{M}_n^\top\,\boldsymbol{R}_O \begin{bmatrix} \boldsymbol{R}_2\,\boldsymbol{R}_1 & 0 & \boldsymbol{R}_2\,\mathbf{t}_1+\mathbf{t}_2 \\ -2\alpha(\mathbf{t}_2^\top\,\boldsymbol{R}_2 + \mathbf{t}_1^\top)\,\boldsymbol{R}_1 & 1 & -\alpha(2\,\mathbf{t}_2^\top\,\boldsymbol{R}_2\,\mathbf{t}_1 +\|\boldsymbol{R}_2\,\mathbf{t}_1\|^2 + \|\mathbf{t}_2\|^2) \\ \mathbf{0}^\top & 0 & 1 \end{bmatrix} \boldsymbol{R}_O^\top\,\boldsymbol{M}_n$$

$$= \boldsymbol{M}_n^\top\,\boldsymbol{R}_O \begin{bmatrix} \boldsymbol{R}_2\,\boldsymbol{R}_1 & 0 & \boldsymbol{R}_2\,\mathbf{t}_1+\mathbf{t}_2 \\ -2\alpha(\mathbf{t}_2^\top + \mathbf{t}_1^\top\,\boldsymbol{R}_2^\top)\,\boldsymbol{R}_2\,\boldsymbol{R}_1 & 1 & -\alpha\|\boldsymbol{R}_2\,\mathbf{t}_1+\mathbf{t}_2\|^2 \\ \mathbf{0}^\top & 0 & 1 \end{bmatrix} \boldsymbol{R}_O^\top\,\boldsymbol{M}_n$$

$$= \boldsymbol{Q}(\boldsymbol{R}_2\,\boldsymbol{R}_1,\ \boldsymbol{R}_2\,\mathbf{t}_1+\mathbf{t}_2). \hspace{6cm} \square$$

$$(71)$$

