# OpenReview forum: "E$(n)$-Equivariant Spherical Decision Surfaces"
_ICLR.cc/2026/Workshop/GRaM — ICLR 2026 Workshop GRaM Poster_

### Official Review · Reviewer_1f8D · 2026-02-20
**Review of E(n)-Equivariant Spherical Decision Surfaces**

**Rating:** 7
**Confidence:** 2

**Review:**

Summary: The paper shows how to construct $E(n)$-equivariant spherical decision surfaces through including translations. They extend prior work on $O(n)$-equivariant hypersphere neurons and show that the resulting decision surfaces are exactly $E(n)$-equivariant.

Strengths:
- Deriving an $E(n)$-equivariant spherical decision surface seems a useful theoretical contribution.
- The paper has extensive background for the unfamiliar reader (like me).
- The decomposition into irreps is useful to connect with pre-existing equivariance research.

Weaknesses:
- I found the framing of the paper to be a bit unclear. It would be useful to explicitly state how this work connects to other equivariant models (e.g. tensor field networks/steerable CNNs) and to emphasize further where this contribution could be used.
- I'm not sure it's feasible, but it would be nice if the main part of the paper could have some sort of explanatory geometric figure (I think this would be useful for the reader).
- I understand that the contribution is theoretical but would it be possible to consider harder datasets than Tetris? If not, there could be more discussion of what would be required to make the approach practical.

Overall, I recommend acceptance as I believe this is an interesting theoretical contribution that will be useful to the GRAM workshop.

**Pmlr Suitability:**

Yes

---

### Official Review · Reviewer_YH9N · 2026-02-21
**Review of E(n) Equivariant Spherical Decision Surfaces**

**Rating:** 6
**Confidence:** 4

**Review:**

# Summary
This paper explores the construction of spherical decision surfaces which are $E(n)$ equivariant. They build upon prior work on $O(n)$ equivariant spherical neurons and add a dimension to include translation equivariance. They prove that this construction satisfies the necessary equivariance constraints and explicitly derive the corresponding group representation of the output. The authors then provide a list of invariants which can be constructed given that inner products no longer work. Finally, they provide some simple experimental tests.

# Strengths and weaknesses
## Strengths
The results are interesting and rigorous. The main results are presented in a logical and clear manner. The decomposition into irreps is useful for connecting to more traditional methods.

## Weaknesses
* It is not clear to me what benefit the spherical neurons allow over more traditional equivariant approaches. Ultimately, the resulting operation from the spherical neurons is just a simple rescaling of the vector component and some combinations of squared distances from the origin but now written in an inconvenient basis (equation 35). However, making this connection explicit appears to be novel even in the $O(n)$ case.
* The background is presented in a disorganized and confusing way. A few sentences to explain the organization of each section and the motivation for each technical part would vastly improve readability. I’ve included some suggestions for how to improve this.
* I encourage the authors to reconsider the notational abuse. While it is readable, changing to $\tilde{\mathbf{R}}$ or something similar would improve clarity.
* It would be helpful to include a roadmap of the structure of the paper in the introduction.
* The experiments are rather limited and do not compare to other equivariant architectures.

### Suggestions for Section 2
The first paragraph does not mention all subsection topics. In particular, spherical neurons are mentioned first but only introduced halfway through.
#### Section 2.1
There is no mention of simplices in the introductory paragraph nor motivation for their introduction. From my understanding, the crux is that in nD, we need $n+1$ points so that any $x\in\mathbb{R}^n$ can be uniquely identified (triangulated) by the squared distances to those $n+1$ points. The regular simplex happens to be a very convenient choice admitting a nice change of basis M (line 071) which conveniently gives back our original point and distance to origin (plus constants). Adding this kind of explanation (and even lower dimensional examples) would make the motivation much clearer.

#### Section 2.2
It makes sense that this is introduced but please also mention it in the first paragraph (lines 045-047)

#### Section 2.3
I would suggest making this the first subsection. I also suggest placing something akin to 117-119 right after line 107 to more precisely define spherical decision surfaces.

#### Section 2.4
Adding some motivation would be helpful. The definition (line 155) is trying to give a canonical way to place a regular simplex with the first vertex being where the decision surface S is.

## Minor issues
* 033 Formatting, use bullet points instead of em-dashes and capitalize the first letter
* 057 and 063 inconsistent indexing
* 210 “it” is ambiguous pronoun

# Questions
* 162-163 is this unique? A reflection is not a rotation?
* 289 How bad is this degeneracy? Of course now there is no unique way to specify a tetrahedron but this is fine since we get the same result no matter the choice? Is the issue that we are only left with scalars in this case?

# Conclusion
Ultimately, I believe there is enough interesting content for acceptance. However, I strongly encourage the authors to improve the organization and writing of the paper.

**Pmlr Suitability:**

Yes

---

### Official Review · Reviewer_mk6s · 2026-02-23
**Strong and relevant theory on exact E(n)-equivariant spherical primitives**

**Rating:** 7
**Confidence:** 3

**Review:**

Relevance: The paper is highly relevant to GRaM because it directly addresses Euclidean-group E(n) equivariance and derives explicit representations for rotations/reflections and translations. It also fits scale and simplicity via a focus on closed-form geometric primitives rather than large architectures.

Novelty: The core novelty is a constructive extension from O(n)-equivariant hypersphere neurons to exact E(n)-equivariance without relying on common engineering workarounds like input centering or explicit pairwise coordinate differences. The decomposition into irreps and the explicit closed-form transformation operators are valuable as a reference. However, the conceptual novelty is incremental relative to the prior spherical-neuron line of work it builds on.

Soundness: The paper checks representation properties and reports numerical verification. It also acknowledges the degenerate case and provides a practical mitigation.

Clarity: The paper is well structured for a theoretical contribution, with clear definitions, explicit equation references, and a helpful structural implications section.

Overall Assessment: This is a technically solid and highly relevant GRaM submission.

**Pmlr Suitability:**

Yes

---

### Meta-Review · Area_Chair_1Vn1 · 2026-02-25

**Decision:**

Accept

**Metareview:**

We are accepting this paper. It's a nice theoretical extension of O(n)-equivariant hypersphere neurons to include translations (E(n)-transformations) explicitly. It's mathematically tidy, doesn't rely on weird hacks, and fits perfectly with the workshop's theme.

**Relevance To Proceedings:**

Yes — suitable for PMLR (long paper)

**Relevance To Workshop:**

Yes — suitable for GRaM

---

### Decision · Program_Chairs · 2026-03-02

Accept (Poster)